# Learning H-Infinity Locomotion Control

**Junfeng Long**[1,*], **Wenye Yu**[1,2,*], **Quanyi Li**[1,*], **Zirui Wang**[1,3], **Dahua Lin**[1,4], **Jiangmiao Pang**[1,†]

[1]Shanghai AI Laboratory [2]Shanghai Jiao Tong University [3]Zhejiang University [4]The Chinese University of Hong Kong

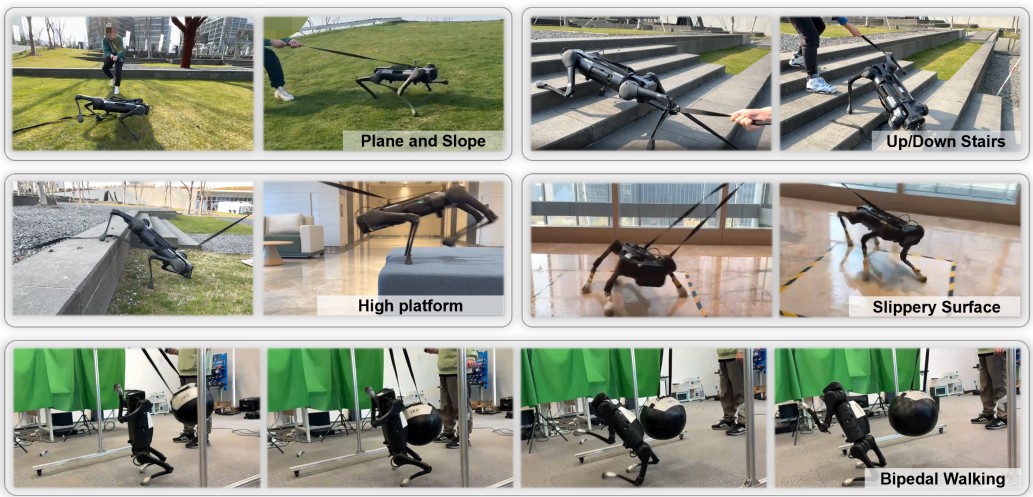

Figure 1: We deploy the policy trained by our method to real robots. Whether in quadrupedal or bipedal states, the robots successfully resist disturbances under various conditions.

**Abstract:**

Stable locomotion in precipitous environments is an essential task for quadruped robots, requiring the ability to resist various external disturbances. Recent neural policies enhance robustness against disturbances by learning to resist external forces sampled from a fixed distribution in the simulated environment. However, the force generation process doesn't consider the robot's current state, making it difficult to identify the most effective direction and magnitude that can push the robot to the most unstable but recoverable state. Thus, challenging cases in the buffer are insufficient to optimize robustness. In this paper, we propose to model the robust locomotion learning process as an adversarial interaction between the locomotion policy and a learnable disturbance that is conditioned on the robot state to generate appropriate external forces. To make the joint optimization stable, our novel $H_{\infty}$ constraint mandates the bound of the ratio between the cost (a measurement for performance drop) and the intensity of the external forces. We verify the robustness of our approach in both simulated environments and real-world deployment, on quadrupedal locomotion tasks and a more challenging task where the quadruped performs locomotion merely on hind legs. Please refer to our project page for videos of real-world deployment.

## 1   Introduction

Recent end-to-end learning-based quadruped controllers exhibit various capabilities during deployment in real-world settings [1, 2, 3, 4, 5, 6, 7, 8, 9, 10, 11, 12, 13, 14, 15, 16]. Moreover, the learning-based approach enables skills beyond locomotion including target tracking in a bipedal manner [17, 18], manipulation using front legs [19], jumping over obstacles [17] and parkour [20].

8th Conference on Robot Learning (CoRL 2024), Munich, Germany.

Successful real-world deployment requires the control policy to be able to resist various disturbances like strong wind and falling debris. Previous learning-based controllers acquire this ability with domain randomization [21, 22] where environment parameters like external forces [23, 24] are randomly sampled and exerted on the robot trunk during training. However, this method is not efficient enough to generate high-quality disturbance-resisting training samples and hinders the policy from acquiring adequate robustness. To be specific, excessively severe disturbances in early training procedures could undermine the training, whereas insufficiently challenging disturbances in late training stages may hinder the robot from developing a more resilient policy. The preliminary experiments in Appendix A provide evidence of this hypothesis.

For generating more effective training samples, an ideal external force sampler is supposed to affect the policy to the extent that the agent experiences an obvious performance drop but is still able to recover from the disturbance, which guarantees not only the training feasibility but the weakness of the policy is attacked precisely. To this end, we introduce a disturber network conditioned on the current states of the robot to generate adaptive external forces. Compared to the actor that aims to maximize the cumulative discounted overall reward, the disturber is modeled as a separate learnable module to maximize the cumulative discounted error between the task reward and its upper bound. To ensure stable optimization between the actor and the disturber, we implement an additional learning objective derived from the constraint inspired by the classical $H_\infty$ theory [25, 26, 27], which mandates the bound of the ratio between the cost and the intensity of external forces generated by the disturber. Following this constraint, we naturally derive an upper bound for the cost function with respect to a certain intensity of external forces, which is equivalent to a performance lower bound for the actor with a theoretical guarantee.

We train our method in Isaac Gym simulator [28] and utilize dual gradient descent method [29] for joint optimization. We evaluate our locomotion policy by comparing it against baseline approaches in terms of their command-tracking ability under various types of disturbances and terrains. We also train policies with baseline methods and our method in the non-stationary bipedal walking setting and measure their abilities to resist collision. In all evaluations, our method outperforms the baseline method, suggesting the effectiveness and superiority of our method. **We deploy the learned policy on Unitree Aliengo robot and Unitree A1 robot in real-world settings. As shown in Fig. 1, the robot manages to traverse planes, slopes, stairs, high platforms, and greasy surfaces whether the external force is applied to the trunk or legs. The robot can even walk with its hind legs while withstanding the impact from heavy objects.**

## 2   Related Work

Quadruped robots are expected to stabilize themselves in face of noisy observations and external forces. While large quantities of research have been carried out to resolve the former issue either by modeling observation noises explicitly during training procedure [9, 24] or introducing visual inputs by depth images to robots [3, 20], few works shed light on confronting potential physical interruptions. While some works claim to achieve robust performance during real-world deployment [7], they fail to model external forces as learnable modules and introduce extreme disruptions to either training or real-world deployment, resulting in vulnerability to harsher conditions.

However, simply modeling external forces as a learnable module causes the problem to fall into the setting of adversarial reinforcement learning, which is a particular case of multi-agent reinforcement learning. One critical challenge in this field is training instability. During training, each agent's policy changes over time, which results in the environment becoming non-stationary from the view of any individual agent. Directly applying single-agent algorithm will lead to the non-stationary problem. For example, Lowe et al. [30] found that the variance of the policy gradient grows exponentially when the number of agents increases. Although following works utilized centralized critic [30, 31] which can stabilize training, the learned policy may be sensitive to its training parameters and converge to a poor local optimal. This problem is more severe for competitive environments because if the opponents change their policies, the learned policy may perform even worse [32].

There exist some previous works on adversarial reinforcement learning aiming to enhance robustness of control policy. For instance, Gleave et al. [33] constructed an adversarial agent that produced effective attack on the actor, but did not proceed to refine the actor for more robust performance. Pinto et al. [34] and Pan et al. [35] drew inspiration from H-infinity theory and adopted an adversarial RL framework. However, the former modeled the problem as a zero-sum game while our method does not mandate this setting. The latter averted risk by lowering the variance of multiple value functions while our approach introduces explicit estimation of performance drop through cost value networks. Besides, they both failed to regulate the ratio between output error and disturbance norm as stated in H-infinity theory, granting no performance lower bound for the actor in face of disturbances. Similar to [34], Rigter et al. [36] modeled a zero-sum game for model-based offline RL. Besides, they constructed worst-case scenario for the actor by tuning the model to predict transitions with the most unpromising next-state value, but an explicit disturbance was absent.

In light of that, we introduce a novel training framework for quadruped locomotion by modeling an external disturber explicitly, which is the first attempt to do so as far as we are concerned. Based on the classic $H_\infty$ method from control theory [25, 26, 27], we devise a brand-new training pipeline where the external disturber and the actor of the robot can be jointly optimized in an adversarial manner. With more experience of physical disturbance in training, quadruped robots acquire more robustness against external forces in real-world deployment.

## 3 Preliminaries

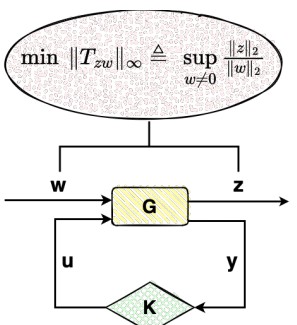

Figure 2: Illustration of the classic $H_\infty$ control theory.

Classic $H_\infty$ control [37] deals with a system where disturbance is involved. We denote $G$ as the plant, $K$ as the controller, $u$ as the control input, $y$ as the measurement available to the controller, $w$ as an unknown disturbance, and $z$ as the error output which is expected to be minimized. In general, we wish the controller to stabilize the closed-loop system based on a model of the plant $G$. As shown in Fig. 2, the goal of $H_\infty$ control is to design a controller $K$ that minimizes the error $z$ while minimizing the $H_\infty$ norm of the closed-loop transfer function $T_{zw}$ from the disturbance $w$ to the error $z$.

However, minimizing $\|T_{zw}\|_\infty$ is usually challenging. In practice, we instead wish to find an acceptable $\eta > 0$ and a controller $K$ satisfying $\|T_{zw}\|_\infty < \eta$, which is called suboptimal $H_\infty$ control. We denote this type of controller as $\eta$-optimal in this paper. According to Morimoto and Doya [25], if $\|T_{zw}\|_\infty < \eta$, it is guaranteed that the system will remain stabilized for any disturbance mapping $\mathbf{d} : z \mapsto w$ with $\|\mathbf{d}\|_\infty < \frac{1}{\eta}$.

Finding a $\eta$-optimal $H_\infty$ controller is modeled as a min-max problem. We consider a plant $G$ with dynamics given by

$$\dot{\mathbf{x}} = f(\mathbf{x}, \mathbf{u}, \mathbf{w}),$$

where $\mathbf{x} \in X \subset \mathbf{R}^n$ is the state, $\mathbf{u} \in U \subset \mathbf{R}^m$ is the control input, and $\mathbf{w} \in W \subset \mathbf{R}^l$ is the disturbance input. Then, the $H_\infty$ control problem can be viewed as finding a controller that satisfies:

$$\|T_{zw}\|_\infty^2 = \sup_{\mathbf{w}} \frac{\|\mathbf{z}\|_2^2}{\|\mathbf{w}\|_2^2} < \eta^2, \tag{1}$$

where $\mathbf{z}$ is the error output. Given that the Euclidean norms $\|\mathbf{z}\|_2$ and $\|\mathbf{w}\|_2$ are defined as:

$$\|\mathbf{z}\|_2^2 = \int_0^\infty \mathbf{z}^T(t)\mathbf{z}(t)\,dt, \quad \|\mathbf{w}\|_2^2 = \int_0^\infty \mathbf{w}^T(t)\mathbf{w}(t)\,dt$$

Our goal, hereby, is to find a control input $\mathbf{u}$ satisfying:

$$V = \int_0^\infty (\mathbf{z}^T(t)\mathbf{z}(t) - \eta^2\mathbf{w}^T(t)\mathbf{w}(t))dt < 0, \tag{2}$$

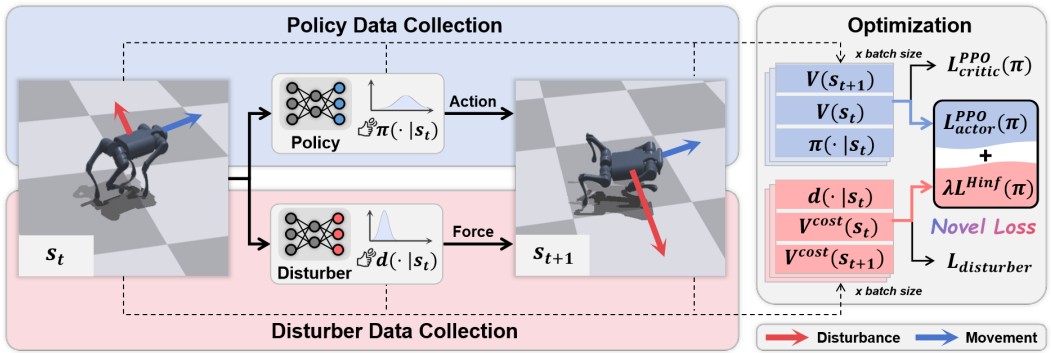

Figure 3: Overview of $H_\infty$ locomotion control method. At every time step during the training process, we perform a simulation step based on the robot's action and the external force generated by the disturber. The agent thus moves towards the rewarded direction and resists the disturbance. During the optimization process, values are calculated for batched training samples and carry out $H_\infty$ policy gradient by optimizing the PPO loss of the actor while taking into consideration the novel constraint $L^{Hinf}$. Value estimators (Critic) are also updated to approximate the state value.

where $\mathbf{w}$ is any possible disturbance and the state vector $\mathbf{x}$ satisfies zero initialization ($\mathbf{x}(0) = \mathbf{0}$). By solving the following min-max game, we can find the best control input $\mathbf{u}$ while the worst disturbance $\mathbf{w}$ is chosen to maximize $V$:

$$V^* = \min_{\mathbf{u}}\max_{\mathbf{w}} \int_0^\infty (\mathbf{z}^T(t)\mathbf{z}(t) - \eta^2\mathbf{w}^T(t)\mathbf{w}(t))dt < 0. \tag{3}$$

# 4 Learning $H_\infty$ Locomotion Control

In this section, we first give the definition of the robust locomotion problem. After that, we describe our method in detail and give a practical implementation.

## 4.1 Problem Definition

As described in the former sections, we wish the disturber to learn more effective disturbances. We model it as a one-step decision problem. Given a Markov Decision Process (MDP) $\mathcal{M} = \{S, A, T, R, \gamma\}$, we define the disturbance policy to be a function $\mathbf{d} : \mathbf{S} \to \mathbf{D} \subset \mathbf{R}^3$, which maps observations to forces. Let $\mathbf{C} : \mathbf{S} \times \mathbf{A} \times \mathbf{D} \to \mathbf{R}^+$ be a cost function that measures the gap between current performance and the best performance in theory measured by reward.

Additionally, $\mathbf{C}_\pi^\mathbf{d}(s) \equiv \mathbb{E}_{(a,d)\sim(\pi(s),\mathbf{d}(s))}\mathbf{C}(s, a, d)$ denotes the gap between expected performance and actual performance given policy $\pi$ and disturber $\mathbf{d}$. Similar to Sec. 3, for a given $\eta > 0$, we wish to find an admissible policy $\pi$ such that

$$\lim_{T\to\infty} \sum_{t=0}^T \mathbb{E}_{s_t}(\mathbf{C}_\pi^\mathbf{d}(s_t) - \eta\|\mathbf{d}_t\|_2) < 0, \tag{4}$$

We define a policy $\pi$ satisfying the above condition as $\eta$-optimal. More intuitively, if a policy is $\eta$-optimal, then an external force $f$ can get a performance decay up to $\eta\|f\|_2$. Additionally, we wish the disturbances to be effective, which means that it can maximize the cost of policy with limited intensity. Therefore, for a policy $\pi$, and a discount factor $0 \leqslant \gamma_2 < 1$, the target of $\mathbf{d}$ is to maximize:

$$\mathbb{E}_\mathbf{d}[\sum_{t=0}^\infty \gamma_2^t(\mathbf{C}_\pi^\mathbf{d}(s_t) - \eta\|\mathbf{d}_t\|_2)] \tag{5}$$

## 4.2 Method

In reinforcement learning-based locomotion control, the reward functions are usually complicated [5, 23, 1, 7]. Some of them guide the policy to complete the task, and some of them act as regularization to the policy. In our work, we divide the reward functions into two categories, the task rewards and the auxiliary rewards. The former part leads the policy to achieve command tracking, maintain good orientation and stay at desired base height, while the latter part leads the policy to satisfy the physical constraints of robot and give smoother control. We present the details of our reward functions in Table 1 and 2, which can be found in Appendix C.1.

Now we denote the rewards from each part as task rewards $R^{task}$ and auxiliary rewards $R^{aux}$ respectively, and the overall reward as $R$. Firstly, we assume that the task reward has an upper bound $R^{task}_{max}$, and the cost can be formulated as $\mathbf{C} = R^{task}_{max} - R^{task}$. With $R$ and $C$, we can get value functions for overall reward and cost, denoted as $V$ and $V^{cost}$. We adopt PPO [38] as our basic policy optimization method. Then the goal of the actor at each iteration is to solve:

$$
\begin{aligned}
\underset{\pi}{\text{maximize}} \quad & \mathbb{E}_t\left[\frac{\pi(a_t|s_t)}{\pi_{\text{old}}(a_t|s_t)}A(s_t)\right] \\
\text{subject to} \quad & \mathbb{E}_t\left[\text{KL}\left[\pi_{\text{old}}\left(\cdot \mid s_t\right), \pi\left(\cdot \mid s_t\right)\right]\right] \leqslant \delta \\
& \mathbb{E}_t\left[\eta\|\mathbf{d}_t\|_2 - \mathbf{C}^{\mathbf{d}}_\pi(s_t)\right] > 0,
\end{aligned}
\tag{6}
$$

where $A$ is the advantage function [39]. As PPO [38] is used to optimize the disturber as well, the goal of the disturber at each iteration is to solve:

$$
\begin{aligned}
\underset{\mathbf{d}}{\text{maximize}} \quad & \mathbb{E}_t\left[\frac{\mathbf{d}(d_t|s_t)}{\mathbf{d}_{\text{old}}(d_t|s_t)}A^{cost}(s_t)\right] \\
\text{subject to} \quad & \mathbb{E}_t\left[\text{KL}\left[\mathbf{d}_{\text{old}}\left(\cdot \mid s_t\right), \mathbf{d}\left(\cdot \mid s_t\right)\right]\right] \leqslant \delta,
\end{aligned}
\tag{7}
$$

However, requiring a high-frequency controller to be strictly robust in every time step is unpractical, so we replace the constraint $\mathbb{E}_t\left[\eta\|\mathbf{d}(s_t)\|_2 - \mathbf{C}^{\mathbf{d}}_\pi(s_t)\right] > 0$ with a more flexible substitute:

$$
\mathbb{E}_t\left[\eta\|\mathbf{d}_t\|_2 - \mathbf{C}^{\mathbf{d}}_\pi(s_t) + V^{cost}(s_t) - V^{cost}(s_{t+1})\right] > 0,
\tag{8}
$$

where $V^{cost}$ is the value function of the disturber. Intuitively, if the policy guides the robot to a better state, the constraint will be slackened, otherwise the constraint will be tightened. We will show that using this constraint, the actor is also guaranteed to be $\eta$-optimal.

We follow PPO to deal with the KL divergence part and use dual gradient decent method [29] to deal with the extra constraint, denoted as $L^{Hinf}(\pi) \triangleq \mathbb{E}_t[\eta\|\mathbf{d}_t\|_2 - \mathbf{C}^{\mathbf{d}}_\pi(s_t) + V^{cost}(s_t) - V^{cost}(s_{t+1})] > 0$, then the update process of policy can be described as:

$$
\begin{aligned}
\pi &= \underset{\pi}{\text{argmax}}L^{PPO}_{actor}(\pi) + \lambda * L^{Hinf}(\pi) \\
\mathbf{d} &= \underset{\mathbf{d}}{\text{argmax}}L_{disturber}(\mathbf{d}) \\
\lambda &= \lambda - \alpha * L^{Hinf}(\pi),
\end{aligned}
\tag{9}
$$

where $L^{PPO}_{actor}(\pi)$ is the PPO objective function for the actor, $L_{disturber}(\mathbf{d})$ is the objective function for disturber with a similar form as PPO objective function, $\lambda$ is the Lagrangian multiplier of the proposed constraint, and $\alpha$ is the step-size of updating $\lambda$. An overview of our method is in Fig. 3.

## 4.3 $\eta$-optimality

We assume that $0 \leqslant \mathbf{C}(s, a) \leqslant C_{max}$ where $C_{max} < \infty$ is a constant, and there is a value function $V^{cost}_\pi$ such that $0 \leqslant V^{cost}_\pi(s) \leqslant V^{cost}_{max}$ for any $s \in \mathbf{S}$, where $V^{cost}_{max} < \infty$. Besides, we denote $\beta^t_\pi(s) = P(s_t = s|s_0, \pi)$, where $s_0$ is sampled from initial states, assuming that the limit of distribution under policy $\pi$ is $\beta_\pi(s) = \lim_{t\to\infty}\beta^t_\pi(s)$ and it exists. Then we have the following theorem:

**Theorem 1.** *If* $\mathbf{C}^{\mathbf{d}}_\pi(s) - \eta\|\mathbf{d}(s)\|_2 < \mathbb{E}_{s'\sim P(\cdot|\pi,s)}(V^{cost}_\pi(s) - V^{cost}_\pi(s'))$ *for* $s \in \mathbf{S}$ *with* $\beta_\pi(s) > 0$, *the policy* $\pi$ *is* $\eta$-*optimal.*

Detailed derivation of Theorem 1 can be found in Appendix B.

## 4.4 Practical Implementations

We use Isaac Gym [28, 40] with 4096 parallel environments and a rollout length of 100 time steps. Our training platform is RTX 3090. During training, we randomize ground friction, restitution coefficients, motor strength, joint-level PD gains, system delay and initial joint positions in each episode. The randomization ranges for each parameter are detailed in Table 3 in Appendix C.2. The algorithm is summarized in Algorithm 1 in Appendix C.3.

## 5 Experimental Results

In this section, we conduct experiments to show the effectiveness of our method. We use the latest non-visual locomotion method [9] as our baseline which is trained with continuous stochastic disturbances drawn from a uniform distribution. By changing the disturbance sampling strategy to ours and its ablated versions, we can show to what extent our method exceeds the baseline and the effectiveness of specific modules of our methods. Our experiments aim to answer these questions:

1. Can our method and its variants handle continuous disturbances as well as the baseline?
2. Can all methods handle the challenges of sudden extreme disturbances?
3. Can all methods resist deliberate disturbances that intentionally attack the policy?
4. Is our method applicable to other tasks that require stronger robustness?
5. Can our method be deployed to real robots?

Specifically, we design four different training settings for comparison studies. First, we train a policy in complete settings where both H-infinity loss and a disturber network are exploited, which we refer to as *ours*. We clip the external forces to have an intensity of no more than 100N for sake of robot capability. Next, we remove the H-infinity loss from the training pipeline and obtain another policy, which we refer to as *ours without hinf loss*. Then, we keep the H-infinity loss but remove the disturber network from *ours* and replace it with a [1]disturbance curriculum whose largest intensity grows linearly from 0N to 100N with the training process and whose direction is sampled uniformly. We call this policy *ours without learnable disturber*. Finally, we train a vanilla policy without both H-infinity loss and disturber network, which also experiences random external forces with curriculum disturbance as described above. We refer to this policy as *baseline*. All four policies are trained on the same set of terrains (Stairs, Slopes, and Discrete heightfield) as is shown in Appendix C.2. The training process for all policies lasts 5000 epochs.

After obtaining the well-trained 4 policies, we evaluate them on 3 terrains with 3 types of disturbances (continuous disturbance, sudden force, and deliberate attack) and measure their command-tracking performance. For each evaluation, we repeat the rollout 32 times with different seeds and report the average performance with a 95% confidence interval.

### 5.1 Can our method and its variants handle continuous disturbances as well as the baseline?

To answer question 1, we test all policies with random continuous disturbances which are drawn from a uniform distribution ranging from 0-100N with the same frequency as controllers. It is the same type of disturbance experienced by the baseline at the final training stage. We command the robot to move forward with a velocity of 1.0 m/s. The tracking curves in Fig. 4 show that our method has the same capability of dealing with continuous disturbances on rough slopes as baseline methods and it even performs better on discrete height fields, and stairs. **In an overall sense, our method can achieve comparable or even better performance against the baseline method in the continuous disturbance setting, even if the baseline methods have been trained with the same type of disturbances. Also, the policy trained without H-infinity loss fails immediately regardless of the terrain, demonstrating that vanilla adversarial training doesn't work well, highlighting the effectiveness of the novel H-infinity loss.**

---

[1]Without the curriculum scheme, the training will collapse as large force may be sampled in the early training phase, which is also confirmed by our preliminary experiments in A.

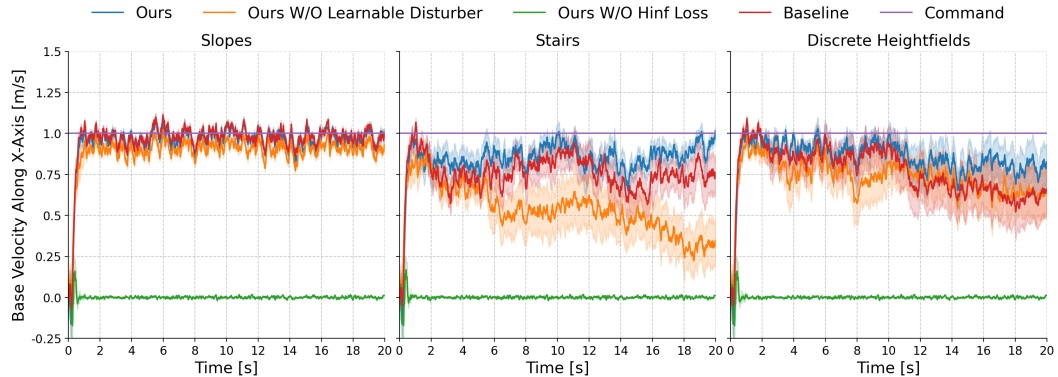

Figure 4: Tracking curve of our method and baselines under continuous random forces.

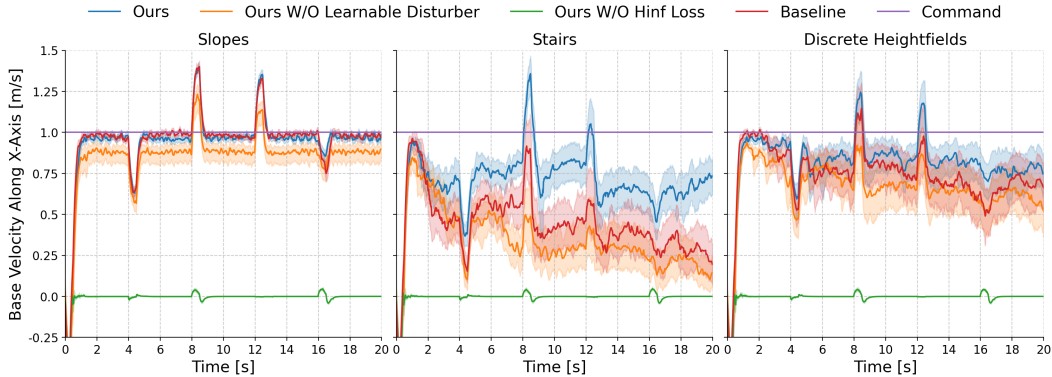

Figure 5: Tracking curve of our method and baselines under sudden large forces.

## 5.2 Can all methods handle the challenges of sudden extreme disturbances?

To answer question 2, we evaluate all policies by applying sudden large external forces on the trunk of robots. We apply identical forces to all robots with an intensity of 150N and a random direction sampled uniformly. The external forces are applied every 4 seconds and last 0.5 seconds. **In Fig. 5, a spike or pit appears at the moment the force is applied, indicating the robot is trying to offset the external force. Robot controlled by our policy shows better precision in tracking the command and the ability to recover from sudden force, especially on stairs and heightfields.**

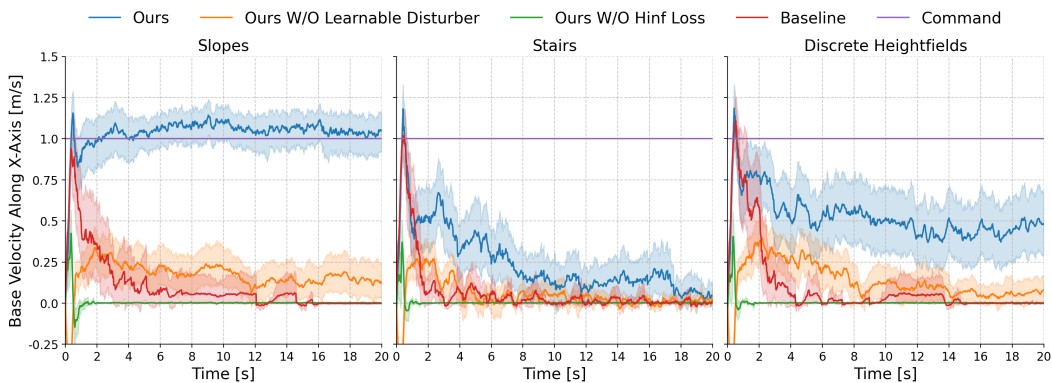

Figure 6: Tracking curve for all methods tested with disturbers trained to intentionally attack them.

### 5.3  Can all methods resist deliberate disturbances that intentionally attack the policy?

To answer question 3, we freeze the parameters of four well-trained policies and train a disturber from scratch for each policy using our method. By doing this, each disturber is optimized to discover the weakness of the corresponding policy and try to undermine its performance as much as possible. We perform the disturber training for 500 epochs and examine the tracking performance of the four policies on different terrains with the specifically trained adversarial disturber. The disturbance are applied continuously as well. **The results shown in Fig. 6 suggest disturbers can identify the weakness for other policies immediately, and these policies fail upon encountering the attach for the first time, whereas our method can withstand the deliberate disturbance many times across three challenging terrains, especially on slopes and discrete heightfields.**

### 5.4  Is our method applicable to other tasks that require stronger robustness?

To answer question 5, we train the robot to walk with two hind legs and test the policy by exerting intermittent large external forces. We train the policy for 10000 epochs for the sake of stronger demands of this task. Identical to the quadrupedal locomotion task, the baseline bipedal policy is trained with a normal random disturber while our method is trained with the proposed adaptive disturber. Both disturbers have the same sample space ranging from 0N to 50N. To evaluate the performance of both

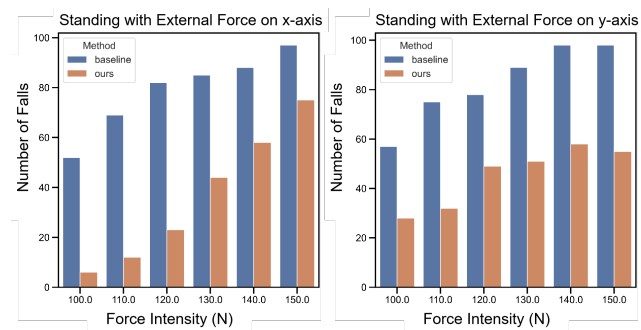

Figure 7: Comparison between the baseline and our method in terms of the number of falls.

methods, we count the total times of falls in one episode when external forces are exerted. Each evaluation episode lasts 20 seconds. Every 5 seconds, the robot receives a large external force with an intensity of 100N to 150N that lasts 0.2 seconds. We carry out two different experiments where the directions of the forces are set to $x, y$ axes respectively. For each method, the evaluation runs 32 times repeatedly and we report the average number of falls. **As shown in Fig. 7, our method outperforms the baseline policy by a large margin no matter the force comes from $x$ or $y$ axes.**

### 5.5  Can our method be deployed to real robots?

To answer question 4, we train our policy with the learnable disturber with a force limit of 100 N and deploy trained policies on Unitree Aliengo quadrupedal robots in the wild. As shown in Fig. 1, **it can traverse various terrains such as staircases, high platforms, slopes, and slippery surfaces, withstand pulling on the trunk, legs, and even arbitrary kicking, and accomplish different tasks such as sprinting.** In addition, We deploy the bipedal walking policy to the Unitree A1 robot. As shown in Fig. 1, **the standing policy is able to withstand collisions with heavy objects and random pushes on its body while retaining a standing posture.**

## 6  Conclusion

In this work, we propose $H_\infty$ learning framework for quadruped locomotion control. Unlike previous works where the external force is drawn from a fixed distribution, we propose to train an adversarial disturber to generate external force dynamically. To stabilize the learning process, we introduce a novel $H_\infty$ constraint to policy optimization, providing a guarantee for the actor's performance lower bound in face of external forces with a certain intensity. We demonstrate our method achieves notable improvement in robustness in both locomotion and standing tasks and can be deployed in real-world settings. To inspire further research, all code and checkpoints are made public.

## Acknowledgments

We thank Shanghai AI Laboratory for hardware support and provision of experimental site.

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

# Appendix

## A Preliminary Experiments

To verify the necessity of having an adaptive disturber, we conduct a preliminary experiment. The most common disturber for legged locomotion randomly samples external forces from $[0, 30]$ N [28]. We increase the upper bound of the uniform distribution to 100 N and get the *Baseline* method. As shown in Fig. 8, the training of *Baseline* collapses under external forces with the extremely large upper bound ($100N$). Although another method, *Baseline-C*, overcomes this by curriculum learning where the upper bound of the forces linearly increases throughout the training, the trained policy fails to achieve comparable final performance against our method, as the training samples generated may be not challenging enough in the late training stage in terms of not only the magnitude but the direction of the force. However, our method keeps optimizing for better adversarial performance and producing valid training samples throughout the training.

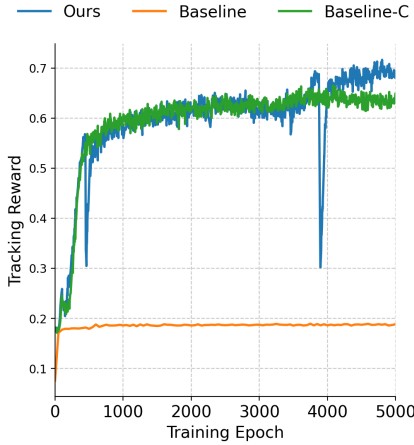

Figure 8: Preliminary experiment comparing policies trained with fixed disturber and our method.

## B Proof of Theorem 1

*Proof.*

$$\lim_{T \to \infty} \frac{1}{T} \sum_{t=0}^{T} \mathbb{E}_{s_t} (\mathbf{C}_\pi^{\mathbf{d}}(s_t) - \eta \|\mathbf{d}_t\|_2)$$

$$< \lim_{T \to \infty} \frac{1}{T} \sum_{t=0}^{T} \mathbb{E}_{s_t} (\mathbb{E}_{s_{t+1} \sim P(\cdot|\pi, s_t)} (V_\pi^{cost}(s_t) - V_\pi^{cost}(s_{t+1}))) \text{ (Hypothesis of the theorem)}$$

$$= \lim_{T \to \infty} \frac{1}{T} \sum_{t=0}^{T} \int_{\mathbf{S}} \beta_\pi^t(s_t)$$

$$\int_{\mathbf{S}} P(s_{t+1}|s_t, \pi)(V_\pi^{cost}(s_t) - V_\pi^{cost}(s_{t+1})) ds_{t+1} ds_t \text{ (Based on the definition of } \beta_\pi^t(s_t))$$

$$= \lim_{T \to \infty} \frac{1}{T} \sum_{t=0}^{T} \int_{\mathbf{S}} \beta_\pi^t(s_t) V_\pi^{cost}(s_t) ds_t$$

$$- \int_{\mathbf{S}} \beta_\pi^t(s_t) \int_{\mathbf{S}} P(s_{t+1}|s_t, \pi) V_\pi^{cost}(s_{t+1}) ds_{t+1} ds_t$$

$$= \lim_{T \to \infty} \frac{1}{T} \sum_{t=0}^{T} (\mathbb{E}_{s_t} V_\pi^{cost}(s_t)$$

$$- \int_{\mathbf{S}} \int_{\mathbf{S}} \beta_\pi^t(s_t) P(s_{t+1}|s_t, \pi) ds_t V_\pi^{cost}(s_{t+1}) ds_{t+1}) \text{(Dominated convergence theorem)}$$

$$= \lim_{T \to \infty} \frac{1}{T} \sum_{t=0}^{T} (\mathbb{E}_{s_t} V_\pi^{cost}(s_t) - \mathbb{E}_{s_{t+1}} V_\pi^{cost}(s_{t+1}))$$

$$= \lim_{T \to \infty} \frac{1}{T} (\mathbb{E}_{s_0} V_\pi^{cost}(s_0) - \mathbb{E}_{s_{T+1}} V_\pi^{cost}(s_{T+1}))$$

$$\leqslant \lim_{T \to \infty} \frac{1}{T} (V_{max}^{cost} - 0) = 0 \text{ (Given } V_{max}^{cost} < \infty)$$

Table 1: Reward functions for Unitree A1 standing task

| Term (* indicates $R^{task}$) | Calculation | Scale |
|---|---|---|
| linear velocity tracking* | $exp(-\|v_{xy} - v_{xy}^{cmd}\|^2/\sigma_{track})\, r_{ori}$ | 1.0 |
| angular velocity tracking* | $exp(-\|\omega_z - \omega_z^{cmd}\|^2/\sigma_{track})\, r_{ori}$ | 0.5 |
| joint velocities | $\|\dot{q}\|^2$ | $-2e^{-4}$ |
| joint accelerations | $\|\ddot{q}\|^2$ | $-2.5e^{-7}$ |
| action rate | $\|a_{t+1} - a_t\|^2$ | $-0.01$ |
| joint position limits | $\mathbb{1}[q \notin (q_{min}, q_{max})]$ | $-10.0$ |
| joint velocity limits | $\mathbb{1}[\dot{q} \notin (\dot{q}_{min}, \dot{q}_{max})]$ | $-10.0$ |
| torque limits | $\mathbb{1}[\tau \notin (\tau_{min}, \tau_{max})]$ | $-10.0$ |
| collision | $\sum_{j \in P} j^{contact}/|P|$ | $-1.0$ |
| extra collision | $\sum_{j \in E_p} j^{contact}/|E_p|$ | $-1.0$ |
| front feet contact | $\mathbb{1}[\sum_{f \in [FL,FR]} f^{contact} == 0]$ | 1.0 |
| orientation $r_{ori}$ | $(0.5 * \cos(v_f \cdot \hat{v}^*) + 0.5)^2$ | 1.0 |
| root height | $\min(e^h, 0.55)$ | 1.0 |

Therefore we obtain $\lim_{T \to \infty} \frac{1}{T} \sum_{t=0}^{T} \mathbb{E}_{s_t}(\mathbf{C}_\pi^\mathbf{d}(s_t) - \eta\|\mathbf{d}_t\|_2) < 0$, and thus, the following inequality is derived:

$$\lim_{T \to \infty} \sum_{t=0}^{T} \mathbb{E}_{s_t}(\mathbf{C}_\pi^\mathbf{d}(s_t) - \eta\|\mathbf{d}_t\|_2) < 0 \qquad (10)$$

## C    Training details

### C.1    Reward function scales for Unitree Aliengo locomotion task and Unitree A1 standing task

Detailed reward functions are shown in Table 1 and Table 2. To clarify the meaning of some symbols used in the reward functions, $P$ denotes the set of all joints whose collisions with the ground are penalized, and $E_p$ denotes the set of joints with stronger penalization. $f^{contact}$ stands for whether foot $f$ has contact with the ground. Moreover, $g$ denotes the projection of gravity onto the local frame of the robot, and $h$ denotes the base height of the robot. In the standing task particularly, we define an ideal orientation $v^*$ for the robot base, which we assign the value $v^* = (0.2, 0.0, 1.0)$, and accordingly define the unit ideal orientation $\hat{v}^* = \frac{v^*}{\|v^*\|}$. We expect the local $x-$axis of the robot, which we denote as $v_f$, to be aligned to $\hat{v}^*$, and thus adopt cosine similarity as a metric for the orientation reward. Besides, we scale the tracking rewards by the orientation reward $r_{ori}$ in the standing task because we expect the robot to stabilize itself in a standing pose before going on to follow tracking commands.

### C.2    Terrains and domain randomization details

We exploit three different types of terrains, slopes, stairs, and discrete height fields during the training procedure, as is presented in Fig. 9. We also introduce terrain curriculum strategy, where the level of terrain difficulty is dynamically adjusted according to the distance that the robot can travel during a fixed duration. Besides, we exploit domain randomization for some simulation parameters, as is shown in Table 3.

Table 2: Reward functions for Unitree Aliengo locomotion task

| Term (* indicates $R^{task}$) | Calculation | Scale |
|---|---|---|
| linear velocity tracking* | $exp(-\|v_{xy} - v_{xy}^{cmd}\|^2/\sigma_{track})$ | 1.0 |
| angular velocity tracking* | $exp(-\|\omega_z - \omega_z^{cmd}\|^2/\sigma_{track})$ | 0.5 |
| $z$-axis linear velocity | $v_z^2$ | $-2.0$ |
| roll-pitch angular velocity | $\|\omega_{xy}\|^2$ | $-0.05$ |
| joint power | $\sum |\tau| \odot |\dot{q}|$ | $-2e^{-5}$ |
| joint power distribution | $\text{Var}[|\tau| \odot |\dot{q}|]$ | $-1e^{-5}$ |
| joint accelerations | $\|\ddot{q}\|^2$ | $-2.5e^{-7}$ |
| action rate | $\|a_t - a_{t-1}\|^2$ | $-0.01$ |
| smoothness | $\|a_t - 2a_{t-1} + a_{t-2}\|^2$ | $-0.01$ |
| joint position limits | $\mathbb{1}[q \notin (q_{min}, q_{max})]$ | $-5.0$ |
| joint velocity limits | $\mathbb{1}[\dot{q} \notin (\dot{q}_{min}, \dot{q}_{max})]$ | $-5.0$ |
| torque limits | $\mathbb{1}[\tau \notin (\tau_{min}, \tau_{max})]$ | $-5.0$ |
| orientation | $\|g_{xy}\|^2$ | $-0.2$ |
| base height | $\|h - h^{target}\|^2$ | $-1.0$ |

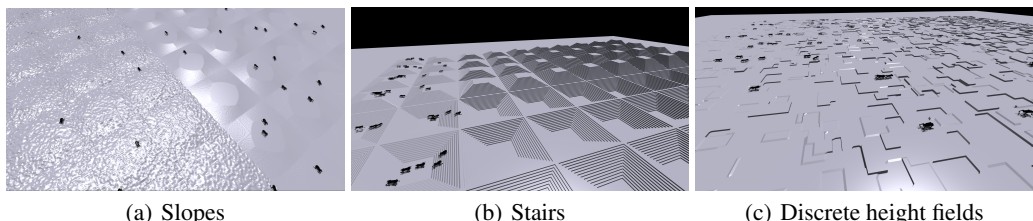

(a) Slopes        (b) Stairs        (c) Discrete height fields

Figure 9: Demonstration of different terrains used in simulated training environments

Table 3: Domain Randomizations and their Respective Range

| Parameters | Range[Min, Max] | Unit |
|---|---|---|
| Ground Friction | $[0.2, 2.75]$ | - |
| Ground Restitution | $[0.0, 1.0]$ | - |
| Joint $K_p$ | $[0.8, 1.2] \times 20$ | - |
| Joint $K_d$ | $[0.8, 1.2] \times 0.5$ | - |
| Initial Joint Positions | $[0.5, 1.5] \times$ nominal value | rad |

### C.3 Pseudo code for $H_\infty$ locomotion control

---

**Algorithm 1:** Learning $H_\infty$ Locomotion Control

---

**Input:** Initial actor $\pi_0$, disturber $\mathbf{d}_0$, overall value function $V_0$, task value function $V_0^{cost}$, initial guess $\eta_0$, initial multiplier $\beta_0$, upper bound of task reward $R_{max}^{cost}$

**Output:** policy $\pi$, disturber $\mathbf{d}$

$\pi_{old} = \pi_0, \mathbf{d}_{old} = \mathbf{d}_0, V_{old} = V_0, V_{old}^{cost} = V_0^{cost}$

**for** iteration = $1, 2, \cdots,$ max iteration **do**

    Run policy $\pi_{old}$ in environment for $T$ time steps

    Compute values of each states with $V_{old}$

    Compute cost values of each states with $V_{old}^{cost}$

    Compute costs $C_t = R_{max}^{task} - R_t$

    Compute advantage estimation $A_t$

    Optimize $\pi$ with $L_{actor}^{PPO} + \lambda * L^{Hinf}$

    Optimize $\mathbf{d}$ with $L_{disturber}$

    $\lambda = \lambda - \alpha * L^{Hinf}$

    $\eta = 0.9 * \eta + 0.1 * \frac{\sum_{t=1}^{T} C_t}{\sum_{t=1}^{T} \|\mathbf{d}_{old}\|_2}$

    $\pi_{old} = \pi$

**end**

---

## D Limitations

Although our method is able to enhance the robustness of control policy across various robots and different tasks, it still suffers from a few limitations. First, task reward functions need to be defined for different embodiments and tasks. Besides, in order to generalize to other robotic systems such as robot arms, our method requires new definitions of "disturbances", other than external force. Future works can be carried out to resolve the aforementioned issues.

