# OpenReview forum: "Learning H-Infinity Locomotion Control"
_robot-learning.org/CoRL/2024/Conference — CoRL 2024_

### Official Review · Reviewer_YAqL · 2024-07-16
**neat idea for constraining policy learning in adversarial scenarios**

**Originality:** 4
**Technical Quality:** 3
**Clarity Of Presentation:** 2
**Potential Impact:** 3
**Recommendation:** 2
**Confidence:** 4

**Review:**

Technical comments:
- “In light of that, we introduce a novel training framework for quadruped locomotion by modeling an external disturber explicitly, which is the first attempt to do so as far as we are concerned.” -> there are numerous other papers which take this perspective of explicit adversarial training in RL. For example: “Robust adversarial reinforcement learning”, “Risk averse robust adversarial reinforcement learning”, “Rambo-rl: Robust adversarial model-based offline reinforcement learning”, “Adversarial policies: Attacking deep reinforcement learning”. Some of these take H_inf perspectives. Authors should make a clear connection to this existing literature, and explain any differences/improvements.
- The development of H_inf preliminaries is unclear to the point of being misleading. For instance: the relationship among the variables is unclear to start with - there is no explicit feedback diagram or algebraic description of how variables relate; it is not clear why (or I believe correct to require) w(0) = 0; the motivation for the game in (4) could be stated more precisely; from (4) it is also straightforward to explain the role of \eta - if \eta is too small, then the disturbance maximizer can just crank up w to force z^2 > \eta^2 w^2 in the limit t \to \infty, and then the problem has no solution. One need not talk about \eta in terms of any kind of “suboptimality.”
- The presentation of (5-7) is confusing because it seems at first as though the object will be solve an H_inf problem with some specific \eta*, however, at (7) we learn that \eta* is never used and the authors instead just pick a (presumably arbitrary) value of \eta. If that is the case, why not skip (5, 6) and just get to (7)?
- (9) and (10) are unclear to me. What are the variables subscripted “old”? I believe them to be the values of those variables at the previous iteration of the (PPO) algorithm; therefore, these equations represent subproblems which that algorithm solves *at every iteration*, rather than the *entire problem* that the algorithms are solving.
- A(s_t) is undefined in (9)
- From the H_inf presentation, I would have expected the method to be solving a zero sum game rather than the general sum game in (9) and (10). In particular, I am not clear on why the authors chose to embed the disturbance constraint the way it is in (9). My best guess is that this has the effect of forcing the policy learner to ignore the objective and focus on being robust to the disturbance when this constraint is violated, and that prevents the policy from learning anything too “non-robust” (i.e., it stabilizes training). It would have been nice to see some kind of (theoretical or experimental) analysis to explain this choice of problem structure.
- The transformation from (8) to (10) also does not make sense: the paper should explain the role of the fraction d/d_old in the objective of (10), which is not there in (8).
- Also, the KL constraints in (9) and (10) should be explained. They are from PPO; nevertheless, this paper will not be self-contained unless there is some explanation of why this is here.
- Vcost is not clearly defined. Assuming that it is a “cost” which the disturber wishes to “minimize” then it seems that an “improvement” at s_{t+1} will actually tighten the constraint. This seems backward. Please take care to explain these details clearly and ensure that all sign conventions are correct.
- More of a nitpick, but Theorem 1 follows immediately from the definition of the modified constraint in (11). I am not really sure why we need a theorem here. It feels like circular logic to me.


Other suggestions for improvement:
- There are numerous grammatical errors. Authors are urged to employ a copy editor or software such as Grammarly.
- The abstract is written in a way that assumes substantial familiarity with low-level details and which will not be accessible to many readers. It is also unclear what the precise innovation (the H_\infty constraint) is doing, in part because “cost” is undefined.
- “Successful real-world deployment requires the control policy to be able to resist various disturbances like strong wind and falling debris. … However, this method is not efficient enough to generate high-quality disturbance-resisting training samples and hinders the policy from acquiring adequate robustness.” -> this is logically inconsistent
- “empirically proved” -> “empirical evidence” is never “proof” (unless it is perhaps providing a counterexample)
- “discounted error between the task reward and its oracle” -> this is unclear. What is the oracle?
- “cost function that measures the errors from commands, expected orientation and base height” -> this is imprecise
- In Fig. 2: “consideration the novel constraint LHinf” -> it appears that this LHinf is actually a cost term, not a constraint. Please take care to use proper terminology.

**Quality Of The Limitations Section:**

1

**Questions For Rebuttal:**

Please see my detailed comments above.

**Robotics Focus:**

4

**Summary Of Paper:**

This paper presents a creative technique for robust RL, which embeds the objective in the traditional zero-sum formulation of an H_inf robust control problem as a constraint during policy learning. Results indicate that this can dramatically improve robustness to a variety of disturbances.

**Summary Of Recommendation:**

The paper contains the kernel of an interesting idea which I believe could have good impact on the community. However, the paper also contains many technical and writing-related issues which should be addressed before publication.

---

### Official Review · Reviewer_ATZL · 2024-07-20
**An impressive and complementary marriage of learning and control, provable in theory and demonstrated in practice.**

**Originality:** 4
**Technical Quality:** 5
**Clarity Of Presentation:** 4
**Potential Impact:** 3
**Recommendation:** 4
**Confidence:** 5

**Review:**

A sound theory with compelling empirical results. The paper proposes an adversarial reinforcement learning inspired by classical $H_{\infty}$ robust control. The proposed choice of the novel $H_{\infty}$ -loss addresses two challenges at once: 1) instability that is often seen in adversarial training and 2) a provable $\eta$-robust controller. The practical considerations taken to relax the constraints to alleviate the conservatism that is often observed in classical robust control are particularly commendable. The rigorous experimentation on the real robot and analysis precisely highlight the key aspects of the approach with relevant ablations over their design choices. The synthesized policies having a competitive performance (at least as good if not better) compared to SOTA heuristic-curriculum methods while being provably robust is impressive. Finally, scaling to complex tasks (the inherently unstable biped handstand), while being robust, motivates the usage of learning-based approaches as an effective alternative to model-based approaches to synthesize robust controllers.

**Quality Of The Limitations Section:**

2

**Questions For Rebuttal:**

The reviewer has no questions or clarifying remarks on the paper.

**Robotics Focus:**

4

**Summary Of Paper:**

A robust control inspired loss for learning robust control policies in adversarial settings.

**Summary Of Recommendation:**

Robust control regularizes learning, while learning addresses the performance trade-off in robust control—an elegant and powerful approach to control synthesis.

---

### Official Review · Reviewer_p1Hn · 2024-07-21
**Nice ideas combining robust classical controls and RL for legged locomotion, overall quality can be further improved.**

**Originality:** 3
**Technical Quality:** 4
**Clarity Of Presentation:** 4
**Potential Impact:** 3
**Recommendation:** 3
**Confidence:** 5

**Review:**

The paper is generally well written and presents an interesting idea taking inspiration from robust control theory to devise a disturbance network that applies the “worst” perturbation force depending on the state of the robot. The authors also present good hardware results.  There are still some avenues for improvement:

**Connection between min-max problem (4) and return for disturber (8):** The connection between H-infinity control and the proposed objective for the disturbance network is not immediately clear. It would be good to add a few remarks in the paper connecting equation (8) and (11) with the H-infinity control equations (1)-(4). This would provide the readers, especially those who do not have a strong background in classical robust control, with a better intuition to the proposed approach.

**Proof for eta-optimality:** Reading through the proof of Theorem 1, it isn’t clear why the policy is eta-optimal as defined in (2). It would be good to add some remarks there connecting the proof to equation (2). Additionally, please provide some text explaining some of the intermediate steps so it is easier for the readers to go through the proof.

**Domain Randomization:** One of the motivations behind this paper is that domain randomization is not very effective in generating high-quality disturbance resisting training samples. Yet, the authors use domain randomization to randomize ground friction and coefficient of restitution. Can we not use the proposed framework to learn an “optimal” disturbance for these parameters as well? If not, it would be good to add a sentence or two in Section 4.4.

**Comparison with Domain Adaptation:** Another approach to deal with sim-to-real gap is domain adaptation where the policy learns to adapt to disturbances online such as in [R1]. It would be good to compare the proposed method, especially in the case where the disturber intentionally attacks the policy (Sec 5.3).

**Limitations of the approach:** This is an important section that is missing from the paper. Please discuss potential limitations and future work.


[R1] Kumar, Ashish, et al. "Rma: Rapid motor adaptation for legged robots." arXiv preprint arXiv:2107.04034 (2021).

**Quality Of The Limitations Section:**

1

**Questions For Rebuttal:**

Please address the points in the Review section.

Particularly, motivate the use of H-infinity control to devise the disturbance network and discuss limitations.

**Robotics Focus:**

4

**Summary Of Paper:**

This paper presents a reinforcement learning framework for quadrupedal robot locomotion that a) learns a disturbance network conditioned on the current state of the robot to maximize the cumulative discounted error in the task performance, and b) learn a policy that can withstand the disturbances applied by the disturbance network. Using ideas from classical robust control theory (H-infinity control), the authors propose a reward function to jointly train the policy and disturbance networks in a stable manner. The proposed approach is validated on two quadrupedal robot platforms under quarupedal and bipedal modes.

**Summary Of Recommendation:**

The method needs to be motivated better, limitations of the approach need to be discussed

---

### Author Rebuttal · Authors · 2024-08-08

We thank all the reviewers and AC for their insightful review and suggestions. We have addressed all their concerns and revised our paper for the readers to better understand our method. Please refer to our revised paper and the rebuttal section for further details.

---

### Decision · Program_Chairs · 2024-09-04

**Decision:**

Accept

**Comment:**

Summary: This paper proposes an adversarial interaction model for robust quadruped locomotion learning, where a learnable, state-conditioned disturbance generates optimal external forces to push the robot to unstable but recoverable states, verified in both simulated and real-world environments, including challenging hind-leg locomotion tasks.

Strength:
* Demonstrates competitive performance on real quadrupedal robots in both quadrupedal and bipedal modes, showing real-world applicability.
* Evaluated on multiple platforms and compared against state-of-the-art methods, showing significant robustness improvements.
* Addresses practical challenges by relaxing constraints to alleviate conservatism typically observed in classical robust control.

Weakness:
* Some sections, such as the connection between H-infinity control and the proposed objective, are not clearly explained.
* Important aspects, such as the detailed explanation of the progressive training and the role of specific parameters, are not well-covered.
* Relies on assumptions like static environments and accurate odometry, which may limit applicability in dynamic real-world scenarios.
* Lacks comprehensive comparison with domain adaptation methods and explicit adversarial training in RL, and does not outperform all baselines.
* The approach appears to rely on heuristics without rigorous proof of correctness, and the uncertainty estimation lacks discussion on false positive and negative rates.

----

Post rebuttal: The paper received two accept and one reject. The AC carefully went throughs the rebuttal and revised paper and discussion and agreed with the majority of the reviewers to accept the paper. But the AC will encourage to improve the clarity of the paper.